# Papain Suppresses Atopic Skin Inflammation through Anti-Inflammatory Activities Using In Vitro and In Vivo Models

**DOI:** 10.3390/antiox13080928

**Published:** 2024-07-30

**Authors:** Hye-Min Kim, Yun-Mi Kang, Minho Lee, Hyo-Jin An

**Affiliations:** 1Department of Oriental Pharmaceutical Science, College of Pharmacy, Kyung Hee University, Seoul 02447, Republic of Korea; mins7576@daum.net; 2Department of Herbology, College of Korean Medicine, Sangji University, Wonju 26339, Republic of Korea; ymkang1013@kiom.re.kr; 3Korean Medicine (KM)-Application Center, Korea Institute of Oriental Medicine (KIOM), 70 Cheomdan-ro, Dong-gu, Daegu 41062, Republic of Korea; 4Department of Life Science, Dongguk University-Seoul, Ilsandong-gu, Goyang-si 10326, Republic of Korea; 5Department of Integrated Drug Development and Natural Products, Graduate School, Kyung Hee University, Seoul 02447, Republic of Korea

**Keywords:** papain, atopic dermatitis, *Dermatophagoides farinae* body, NC/Nga, keratinocytes

## Abstract

Papain (PN) is a proteolytic enzyme derived from *Carica Papaya* L. While the pharmacological effects of PN have not been extensively studied compared to its enzymatic activity, PN also holds potential benefits beyond protein digestion. This study aimed to investigate the potential effects of PN against skin inflammation in house dust mite *Dermatophagoides farinae* body (Dfb)-exposed NC/Nga atopic dermatitis (AD) mice and human HaCaT keratinocytes and their underlying mechanisms. The effects of PN on the skin were assessed via histological examination, measurements of transepidermal water loss (TEWL), quantitative reverse transcription-polymerase chain reaction, Western blotting, and enzyme-linked immunosorbent assay. Our findings indicated that the oral intake of PN decreased the severity scores of lesions resembling AD, TEWL, and the levels of inflammatory cytokines and serum immunoglobulin E in Dfb-induced AD mice, along with a reduction in epidermal thickness and mast cell infiltration. Additionally, PN inhibited the activation of the mitogen-activated protein kinases (MAPKs) and the signal transducer and activator of transcription (STAT) pathways in Dfb-induced AD mice and HaCaT keratinocytes. Moreover, PN improved survival and reduced ROS production in H_2_O_2_-damaged HaCaT keratinocytes and enhanced the expression of antioxidant enzymes in Dfb-induced AD mice. Concludingly, the oral administration of PN suppressed inflammatory mediators and downregulated the MAPKs/STAT pathway, suggesting its potential role in AD pathogenesis.

## 1. Introduction

Atopic dermatitis (AD) represents a persistent inflammatory skin disease with intricate pathophysiological processes. The compromised epidermal barrier function can stimulate the inflammation and infiltration of T cells. Moreover, the activated immune response may exacerbate epidermal barrier dysfunction [1,2]. Prolonged inflammation can cause deleterious changes in the skin tissue, with the severity depending upon the molecular pathways involved in the inflammatory cascade. Furthermore, the mitogen-activated protein kinase (MAPK) and signal transducer and activator of transcription (STAT) pathways represent pivotal mechanisms in the pathogenesis of inflammatory skin disorders [3,4] and are speculated as potential targets for addressing AD [5,6,7].

A considerable proportion of patients with AD exhibit increased sensitivity to house dust mites (HDM), recognized as the major aeroallergen contributing to allergic conditions [8]. Proteases and lipid-binding proteins in HDM allergens have been found to enhance the permeability of the airway epithelium through the cleavage of junctional proteins, thus facilitating allergen access and allergic sensitization [9,10]. This disruption of the barrier function leads to the upregulation of the expression of the pro-allergic mediator thymic stromal lymphopoietin (TSLP), which causes a differentiation of T helper type 2 (Th2) cells and the subsequent secretion of pro-inflammatory cytokines, such as interleukins [11,12]. NC/Nga mice, an inbred strain exhibiting increased susceptibility to HDM, manifest severe dermatitis compared with other strains and show symptoms mirroring those observed in human patients with AD [13]. Consequently, these mice serve as an appropriate model for studying human AD.

Papain (PN), a proteolytic enzyme derived from the latex of the *Carica papaya* L. (papaya) plant, has garnered attention for its traditional use as a remedy to alleviate pain, swelling, and inflammation, as well as to aid digestion [14]. Our previous research has demonstrated the anti-inflammatory properties of PN in obesity-related inflammation [15]. Additionally, papaya extract exhibits anti-skin aging properties attributable to its antioxidant and anti-inflammatory properties [16]. Based on these findings, we posited that PN might have a role in mitigating skin inflammatory conditions induced by HDM. Thus, this study aimed to examine the impact of PN on *Dermatophagoides farinae body* (Dfb)-induced atopic dermatitis (AD) in NC/Nga mice and human HaCaT keratinocytes and to clarify the underlying mechanisms.

## 2. Materials and Methods

### 2.1. Chemicals and Reagents

Papain (PN, derived from papaya latex, cat. no. P4762) and chemicals such as 3-(4,5-dimethylthiazol-2-yl)-2,5-diphenyl tetrazolium bromide (MTT) and dimethyl sulfoxide (DMSO) were sourced from Sigma-Aldrich (EMD Millipore, Billerica, MA, USA). Bio-Techne Ltd. (Abingdon, OX, UK) supplied recombinant human TNF-α and IFN-γ. Dulbecco’s modified Eagle’s medium (DMEM), fetal bovine serum (FBS), penicillin, and streptomycin were acquired from Life Technologies, Inc. (Grand Island, NY, USA). Primary antibodies against p-STAT1, STAT1, p-ERK, ERK, p-JNK, JNK, p-p38, p38, p-STAT6, STAT6, superoxide dismutase (SOD) 2, and nuclear factor erythroid-2-related factor 2 (Nrf2) were obtained from Cell Signaling Technology Inc. (Danvers, MA, USA). Santa Cruz Biotechnology Inc. (Dallas, TX, USA) supplied the primary antibodies against p-IκB-α, heme oxygenase 1 (HO-1), SOD1, glutathione peroxidase (GPx) -4, NAD(P)H quinone oxidoreductase 1 (NQO1), and β-actin. Secondary antibodies conjugated with horseradish peroxidase were procured from Jackson ImmunoResearch Laboratories Inc. (West Grove, PA, USA). Each ELISA kit for IgE, TNF-α, and IL-6 was acquired from R&D Systems Inc. (Minneapolis, MN, USA).

### 2.2. Dfb-Induced AD Model

Male NC/Nga mice (6-week old, 20–25 g weight) were procured from Daehan BioLink (Eumsung, Republic of Korea), a branch of Charles River Japan (Kanagawa, Japan), and housed under controlled conditions: 12 h light/dark cycle, humidity at 40–60%, temperature maintained at 20–25 °C. Mice were randomly divided into five groups (*n* = 6 per group, total of thirty mice): normal group, HDM-induced group, dexamethasone (DEX, positive control) group, and PN oral administration groups (2.5 or 5 mg/kg). AD-like skin lesions were induced by topically applying 100 mg of crude extract from Dfb (Biostir AD; Biostir, Hyogo, Japan) to the shaved dorsal area of mice. Mite antigen was applied twice a week over a six-week period. Barrier disruption was induced through treating with 150 μL of 4% sodium dodecyl sulfate (SDS) three hours prior to Dfb ointment treatment. After the first challenge to induce AD-like symptoms for two weeks, the mice were topically administered vehicle, DEX (5 mg/kg, p.o. dissolved in phosphate-buffered saline [PBS]), or PN (2.5, or 5 mg/kg, p.o. dissolved in PBS) 4 h after Dfb treatment once a day. The same volume of PBS was administered to the normal group as the vehicle. PN concentration was determined according to previous studies [15,17]. Mice were sacrificed at the end of the experiment. Skin samples were collected from the dorsal region of mice and used for histological and Western blot analyses. The experimental procedures followed the university’s guidelines and were approved by the Ethical Committee for Animal Care and Use of Laboratory Animals at Sangji University (Wonju, Korea; approval no. 2020-02).

### 2.3. Dermatitis Severity Score

The severity of clinical dermatitis was evaluated based on the following scoring system. Skin symptoms such as erythema/hemorrhage, scarring/dryness, edema, and excoriation/erosion were scored as follows: 0 for none; 1 for mild (<20%); 2 for moderate (20–60%); and 3 for severe (>60%). [18]. The sum of individual scores was used as the dermatitis severity score.

### 2.4. Measurement of Transepidermal Water Loss

Transepidermal water loss (TEWL) from the dorsal skin of NC/Nga mice was assessed at the end of eight weeks using GPskin Barrier Light (Gpskin, Seoul, Republic of Korea), according to previous reports. TEWL was estimated under specific conditions of 50–55% humidity and 24 °C temperature. At the center of the shaved dorsal area of each mouse, a probe was placed to record the TEWL value in g/m^2^/h. Statistical results were expressed in terms of fold change compared with the control group.

### 2.5. IgE Measurement

Upon completion of the experiment, blood samples were taken from each mouse, and serum was separated via centrifuging at 1700× *g* for 30 min and then stored at −80 °C for future analysis. IgE levels were quantified using an ELISA kit following the manufacturer’s instructions. 

### 2.6. Quantification of Cytokine Levels

The protein extracted from dorsal skin was solubilized in PRO-PREP™ solution (Intron Biotechnology Inc., Seoul, Republic of Korea), and incubated for 20 min at 4 °C. Debris was cleared using microcentrifugation at 12,000× *g* for 30 min at 4 °C, and the supernatant was promptly frozen. Protein concentration was measured using Bio-Rad reagent (Bio-Rad Laboratories, Inc., Hercules, CA, USA) as per the manufacturer’s instructions. TNF-α and IL-6 levels were quantified using ELISA kits, according to the manufacturer’s guidelines.

### 2.7. Histological Analysis and Immunohistochemical Analysis

At the conclusion of the study, the mice’s dorsal skin was fixed in 10% buffered formalin, paraffin-embedded, sectioned to 4 μm thickness, and stained with hematoxylin and eosin (H&E) or toluidine blue to assess epidermal thickness and inflammatory cell infiltration. For immunohistochemistry (IHC), tissue samples were fixed in 4% formaldehyde, paraffin-embedded, and cut into 4 μm sections. The slides were deparaffinized in xylene, rehydrated with graded ethanol, and then hydrated in water. Endogenous peroxidase activity was quenched with 0.6% H_2_O_2_ in 50% MeOH, followed by 0.3% Triton in PBS for permeabilization. The slides were pre-blocked with 10% normal goat serum for one hour and incubated overnight at 4 °C with primary antibody. After washing, the sections were incubated with horseradish peroxidase-conjugated secondary antibodies for one hour at room temperature. Immunoreactivity was visualized with a 3,3-diaminobenzidine chromogen and counterstained with H&E. The morbid alteration was examined using a DM IL LED microscope (Leica Microsystems, Wetzlar, Germany) and photographed with a DFC295 camera (Leica Microsystems). Digital images were captured and analyzed using Leica Application Suite (Leica Microsystems).

### 2.8. RT-qPCR Analysis

The cells or liver tissues were processed to isolate total RNA using the easy blue solution (Intron Biotechnology, Inc., Seoul, Republic of Korea) following the manufacturer’s instructions. Total RNA quantity was assessed via the Epoch microvolume spectrophotometer system (BioTek Instruments, Inc., Winooski, VT, USA). Subsequently, cDNA synthesis was performed with 2 μg of total RNA, d(T)16 primer, and avian myeloblastosis virus reverse transcriptase for a genomic DNA removal step. RT-qPCR (Real Time PCR System 7500; Applied Biosystems, Thermo Fisher Scientific, Inc., Waltham, MA, USA) with SYBR Premix Ex Taq was used to dictate relative gene expression levels. Gene expression fold changes were estimated using the comparative quantification cycle (Cq). The Cq values of target genes were standardized against GAPDH deploying ABI Gene Express 2.0 software (Applied Biosystems, Thermo Fisher Scientific, Inc., Waltham, MA, USA). The primer sequences for real-time reverse transcription-polymerase chain reaction (RT-PCR) were as follows: mouse IL-4 (NM_021283.2), forward 5′-ATCATCGGCATTTTGAACGAGGTC-3′, and reverse 5′-ACCTTGGAAGCCCTACAGACGA-3′; mouse IL-13 (NM_008355.3), forward 5′-AACGGCAGCATGGTATGGAGTG-3′, and reverse 5′-TGGGTCCTGTAGATGGCATTGC-3′; mouse TSLP (NM_021367.2), forward 5′-GCAAATCGAGGACTGTGAGAGC-3′, and reverse 5′-TGAGGGCTTCTCTTGTTCTCCG-3′; mouse IL-17A (NM_010552.3), forward 5′-CAGACTACCTCAACCGTTCCAC-3′, and reverse 5′-TCCAGCTTTCCCTCCGCATTGA-3′; mouse IL-17E (NM_080729.3), forward 5′-TGGCAATGATCGTGGGAACC-3′, and reverse 5′-GAGAGATGGCCCTGCTGTTGA-3′; mouse IL-17F (NM_145856.2), forward 5′-TGCTACTGTTGATGTTGGGAC-3′, and reverse 5′-CAGAAATGCCCTGGTTTTGGT-3′; mouse NRF2 (NM_010902.5), forward 5′-CAGCATAGAGCAGGACATGGAG-3′, and reverse 5′-GAACAGCGGTAGTATCAGCCAG-3′; mouse NQO1 (NM_008706.5), forward 5′-GCCGAACACAAGAAGCTGGAAG-3′, and reverse 5′-GGCAAATCCTGCTACGAGCACT-3′; and mouse GAPDH (NM_001289726.2), forward 5′-GACGGCCGCATCTTCTTGT-3′, and reverse 5′-CACACCGACCTTCACCATTTT-3′.

### 2.9. Cell Viability

To assess PN toxicity in HaCaT keratinocytes, an MTT assay was conducted. Cells were seeded in 96-well culture plates at a density of 5 × 10^4^ cells/mL, allowed to attach for 24 h, and then treated with medium containing different PN concentrations. Following a 24 h incubation period, the cells were exposed to 50 μL of MTT solution (5 μg/mL) for 4 h. Subsequently, the formazan crystals were dissolved in DMSO, and the optical density was measured at 540 nm using a microplate reader.

### 2.10. Cell Culture and Sample Treatment

HaCaT keratinocytes, generously obtained from Professor Jae-Young Um (Kyung Hee University, Republic of Korea), were cultured at 37 °C in DMEM supplemented with 10% FBS, 100 U/mL penicillin, and 100 μg/mL streptomycin under a 5% CO_2_ humidified atmosphere. HaCaT keratinocytes were seeded at a density of 1 × 10^5^ cells per well, starved in 0.1% FBS medium for 24 h, and subsequently treated with PN at concentrations of 100, 200, and 400 μg/mL for 1 h at 37 °C in 5% CO_2_. Following treatment, the cells were stimulated with 10 ng/mL TNF-α/IFN-γ at 37 °C for the specified duration. 

### 2.11. Western Blot Analysis

Samples of cells, liver tissue, and dorsal tissue were homogenized in PRO-PREP™ protein extraction solution (Intron Biotechnology, Inc., Seoul, Republic of Korea) and incubated at 4 °C for 20 min. The homogenates were clarified using centrifugation at 11,000× *g* for 30 min at 4 °C, and the supernatant was immediately frozen. Protein concentrations were determined using Bio-Rad protein assay reagent (Bio-Rad Laboratories, Inc., Hercules, CA, USA) following the manufacturer’s instructions. For protein analysis, samples (10–30 μL) of treated and untreated cell extracts were separated by 8–12% SDS-PAGE and transferred onto polyvinylidene fluoride membranes. The membranes were blocked with 5% skim milk at room temperature for 1 h, then incubated overnight at 4 °C with primary antibodies (1:1000 dilution). Membranes were washed three times with Tween 20/Tris-buffered saline (T/TBS) and then incubated with horseradish peroxidase-conjugated secondary antibody (1:2000 dilution) for 2 h at room temperature. Following three additional washes with T/TBS, protein bands were visualized using intensified chemiluminescence (GE Healthcare Life Sciences, Chalfont, UK). Densitometric analysis was conducted using Bio-Rad Quantity One software version 4.3.0 (Bio-Rad Laboratories, Inc., Hercules, CA, USA). 

### 2.12. Intracellular Reactive Oxygen Species Assay 

The cells were seeded in 96-well culture plates at a density of 2 × 10^5^ cells/mL in culture medium and allowed to attach for 24 h. The cells were then treated with PN in serum-free DMEM. After 24 h, the cells were washed with PBS and treated with 500 µM H_2_O_2_ for 4 h. Following three washes with PBS buffer, the cells were treated with 10 µM DCF-DA (D6883; Sigma-Aldrich) and then incubated in the dark at 37 °C for 20 min. Fluorescence microscopy (Nikon Corporation, Tokyo, Japan) was employed to observe and capture images of the samples, and reactive oxygen species (ROS) levels were quantified using Molecular Devices SoftMax^®^ Pro 6 software. The excitation wavelength used was 488 nm.

### 2.13. Statistical Analysis

Data are presented as the mean ± standard deviation from triplicate experiments. Statistically significant differences were juxtaposed using one-way analysis of variance and Dunnett’s post hoc test. A *p*-value < 0.05 was regarded as statistically significant. SPSS statistical analysis software (version 19.0, IBM SPSS, Armonk, NY, USA) was utilized for the statistical analysis.

## 3. Results

### 3.1. PN Alleviated the Severity of AD-like Skin Lesions and Inflammatory Response in Dfb-Induced AD Mice

To induce AD-like skin inflammation, mice were repeatedly subjected to a Dfb challenge. The details of the experimental approach are depicted in Appendix A. Our observations revealed that mice exposed to Dfb developed AD-like symptoms, including dryness, edema, and excoriation, and showed increased dermatitis scores compared with those of normal, untreated mice. Conversely, treatment with PN led to a reduction in dermatitis scores relative to Dfb-challenged mice, as shown in Figure 1A. Appendix A presents representative images of the dorsal skin of mice in the five experimental groups. The administration of PN significantly lowered TEWL, a crucial indicator of barrier dysfunction (Figure 1B), as well as serum immunoglobulin E (IgE) levels compared with Dfb-exposed mice (Figure 1C). Furthermore, treatment with PN resulted in the downregulation of mRNA expression and the production of the pro-inflammatory cytokines TNF-α and IL-6 (Figure 1D–G). These findings suggest that PN administration mitigates the phenotypes associated with Dfb-induced AD and regulates the production of inflammatory cytokines.

### 3.2. PN Improved Histological Alterations in the Skin of Dfb-Induced AD Mice

To confirm the Dfb-induced histological alterations, we performed H&E and toluidine blue staining which displayed that the Dfb challenge led to hyperkeratosis, evidenced by increased epidermal thickness and inflammatory cell infiltration in the dorsal skin of mice, whereas PN treatment improved histological alterations (Figure 2A,B). PN reversed Dfb-induced epidermal thickening (Figure 2C). Moreover, PN treatment decreased the number of infiltrated mast cells (Figure 2D). 

### 3.3. PN Downregulated AD-Related Cytokines and Upregulated Anti-Oxidative Markers in Dfb-Induced AD Mice

To evaluate whether PN reduces the expression of AD-related cytokines in Dfb-induced AD mice, the mRNA levels of Th2, TSLP, and Th17 cytokines were measured in the dorsal skin of mice. As shown in Figure 3, the mRNA levels of Th2 cytokines (IL-4, IL-13, and IL-5), TSLP, and Th17 cytokines (IL-17A, IL-17E, and IL-17F) increased after Dfb application in AD mice. However, the oral administration of PN downregulated AD-related cytokines in Dfb-induced AD mice. Meanwhile, PN reversed the decrease in the levels of the anti-oxidative markers Nrf2 and NQO1 (Figure 3G,H).

### 3.4. PN Inhibited the Activation of STAT1 and MAPKs in Dfb-Induced AD Mice

To elucidate the potential mechanism underlying the inhibitory action of PN against AD, we assessed the activation of STAT1 and MAPKs using Western blot analysis. The application of Dfb substantially induced the phosphorylation of IκB-α and STAT1; however, the protein levels were notably attenuated using PN in Dfb-induced AD mice (Figure 4A). Furthermore, PN treatment suppressed Dfb-induced MAPKs activation to a level lower than that observed in the DEX group (Figure 4B). These findings suggest that PN treatment exerts an inhibitory effect against AD by suppressing AD-related mediators and downregulating both the STAT1 and MAPKs pathways.

### 3.5. PN Inhibited the NF-κB Signaling Pathway in TNF-α/IFN-γ-Induced HaCaT Keratinocytes

We explored whether PN exhibited a similar mechanism in TNF-α/IFN-γ-stimulated HaCaT human keratinocytes. To establish the optimal concentration of PN, HaCaT cells were exposed to PN concentrations ranging from 7.8 to 125 μg/mL for 24 h. Notably, we observed the cytotoxic effects of PN on HaCaT keratinocytes at the highest concentration (125 μg/mL). Therefore, concentrations of 25, 50, and 100 μg/mL of PN were selected for further in vitro investigations (Figure 5A). Subsequently, we assessed the effect of PN on the expression of HO-1 and observed a significant increase in HO-1 expression (Figure 5B). Next, we investigated the effect of PN on the NF-κB signaling pathway. We found that PN remarkably inhibited the TNF-α/IFN-γ-induced phosphorylation of IκBα and the activation of NF-κB in HaCaT keratinocytes (Figure 5C).

### 3.6. PN Inhibited the Activation of STATs and MAPKs in TNF-α/IFN-γ-Stimulated HaCaT Keratinocytes

Moreover, we assessed the effect of PN on the STAT and MAPKs pathways. The results showed that PN significantly suppressed the TNF-α/IFN-γ-induced phosphorylation of STAT1, STAT6, and the MAPKs ERK and JNK in HaCaT keratinocytes (Figure 6A,B). These results indicate that PN’s inhibitory effect on AD-like skin lesions may result from modulating MAPK phosphorylation and activating NF-κB and STATs in TNF-α/IFN-γ-induced HaCaT keratinocytes.

### 3.7. PN Reduced Oxidative Damage in HaCaT Keratinocytes and Dfb-Induced AD Mice

The effect of PN against intracellular ROS was explored using an oxidative damage model of HaCaT keratinocytes induced using H_2_O_2_. The survival of HaCaT keratinocytes was reduced after H_2_O_2_ treatment to 88.18 ± 0.53%. However, survival increased after treatment with PN, reaching 91.17 ± 2.54%, 94.90 ± 5.87%, and 97.14 ± 3.70% with increasing concentrations of PN (Figure 7A). H_2_O_2_-induced ROS production in HaCaT keratinocytes increased to 115.03 ± 5.98%. This was significantly reduced after PN treatment to 103.37 ± 1.13%, 103.47 ± 1.66%, 103.47 ± 1.66%, and 100.62 ± 1.01% in order of increasing PN concentration (Figure 7B,C). To confirm the potential antioxidant effect of PN, we examined the expression of enzymes related to oxidative stress in an animal model. SOD1 and SOD2 are enzymes known to convert superoxide radicals into hydrogen peroxide, thereby preventing oxidative damage [19,20]. In the Dfb-stimulated group, the expression of these enzymes decreased, whereas it significantly increased in all PN-treated groups, surpassing the levels observed in the DEX group. We assessed the expression of GPx-4, an enzyme known to protect cellular membranes from oxidative damage by eliminating lipid peroxides [21], and found that PN treatment reversed the reduced expression of GPx-4 after Dfb exposure (Figure 7D). Similarly, PN treatment showed a marked increase in the expression levels of NQO1, an enzyme that reduces quinones to hydroquinones, protecting cells from oxidative stress and toxic substances, and HO-1, which degrades heme to exert antioxidant and anti-inflammatory effects [19,22], which were decreased after Dfb stimulation (Figure 7E). In both TNF-α/IFN-γ-treated HaCaT cells and Dfb-induced animal models, HO-1 expression significantly recovered after PN treatment, as evidenced in the epidermal tissue analyzed through IHC. HO-1 expression, indicated via brown staining, was notably reduced in the Dfb-stimulated group compared with the positive control group; however, this increased after PN treatment, with the most pronounced expression in the skin of mice in the PN5 group. In the normal group, brown staining was observed throughout the epidermis, with numerous brown spots in the dermis. In the HDM-stimulated group, no staining was detected in either the epidermis or the dermis. However, in the PN5 group, brown staining was observed in the epidermis, though it was weaker than in the normal group, and in the dermis, the brown staining was much stronger than in the normal group (Figure 7F). Based on the previous findings that the expression of antioxidant-related factors, particularly NQO1 and HO-1, is regulated using Nrf2 [23], we examined the expression of Nrf2. The results showed that Nrf2 expression decreased with Dfb stimulation; however, this significantly recovered with DEX and PN treatment. These findings suggest that PN exhibits protective effects against oxidative stress and oxidative damage in both the in vitro model and the Dfb-induced animal model.

## 4. Discussion

This study highlights the anti-inflammatory properties of PN in both the skin lesions of Dfb-induced AD mice and human epidermal keratinocytes. PN administration mitigated Dfb-induced dermatitis symptoms, including epidermal thickening, mast cell infiltration, TEWL, IgE production, and pro-inflammatory cytokine levels. Our findings elucidate that the anti-inflammatory mechanisms of PN involve the inhibition of the MAPKs and STAT signaling pathways. Intriguingly, while some studies suggest that digestive enzymes can induce acute inflammatory reactions [24,25], our results contradict this notion. PN, a cysteine protease, typically triggers a type 2 response characterized by specific IgE production and the recruitment of inflammatory cells. Furthermore, the intranasal administration of PN has been shown to activate the MAPKs pathway and induce the production of TSLP and IL-25 independently of the IL-1R1/MyD88 signaling in a mouse model of allergic asthma [26]. Epicutaneous sensitization with PN not only disrupted the compromised skin barrier by intensifying TEWL, reaching tight junction proteins, and causing vasodilation [27]; it also influenced Th2 differentiation, skin eosinophilia, IgE, and IgG responses in mice [28,29,30,31,32]. These conflicting perspectives may stem from variations in the route of administration, PN dosage, mouse strain, cell line, and experimental conditions.

Could the oral administration of PN aid in ameliorating skin inflammation? PN exhibits favorable absorption from the gastrointestinal tract and augments the permeability of low-molecular-weight compounds across the small intestinal mucosal barrier [33,34]. Among the various allergens found in wheat grains, gliadins stand out as primary allergens due to the presence of allergenic epitopes in native proteins, which can trigger gluten-sensitive enteropathy, known as celiac disease. Consequently, enzymatic treatment holds promise for reducing food antigenicity by eliminating allergenic epitopes and hydrolyzing proteins into smaller peptides. PN demonstrates significant efficacy in diminishing the allergenic protein gliadin content of wheat flour, resulting in reduced IgE binding [35]. PN has also been employed to mitigate allergies associated with leaky gut syndrome, hypochlorhydria, and intestinal dysbiosis, such as gluten intolerance [36]. Overall, papaya enzymes facilitate digestion by breaking down proteins into amino acids and degrading allergens, thereby alleviating symptoms in patients with diverse digestive disorders and associated allergic conditions. These properties of PN may contribute to alleviating AD-induced skin symptoms.

Oral prescription products containing PN have been granted orphan drug status by the Food and Drug Administration [37]. Despite the widespread use of PN, limited information is available regarding the toxicological and mutagenic properties of PN. Studies conducted by Claudia et al. have suggested that PN does not exhibit toxicity or mutagenicity and it has antioxidant potential against oxidative stress in bacterial systems [38]. Additionally, PN has been demonstrated to possess antioxidant activity, effectively countering H_2_O_2_-induced damage [38]. Furthermore, the antioxidant properties of unripe *Carica papaya* have been shown to safeguard the endothelium against oxidative damage, primarily by reducing intracellular ROS levels, enhancing catalase activity, and inhibiting NF-κB signaling [31]. In this study, we also observed that PN upregulated the expression of the transcription factor Nrf2 that stimulates the transcription of various genes by fastening the antioxidant response element in the promoter regions of target genes [39,40]. As shown in Figure 7D–G, PN significantly restored the expression of SOD1, SOD2, GPx-4, NQO1, and HO-1. Moreover, NQO1, the enzyme implicated in protecting against oxidative stress, was elevated (Figure 3G,H and Figure 7E), along with the antioxidant protein HO-1 (Figure 5B and Figure 7E). The effect of PN may help attenuate skin inflammation by inducing the expression of antioxidant enzymes and transcription factors under skin inflammatory conditions. However, the effects of orally administered PN as an antioxidant have not been thoroughly investigated in the context of allergic disorders. Therefore, research is needed to fully understand these benefits and establish the most effective ways to utilize PN for its antioxidant properties. Moreover, AD is an extremely complex condition. The NC/Nga mouse model used in this study, although appropriate for investigating a specific genetic mutation related to AD, may not entirely replicate human AD. Additionally, the HaCaT keratinocyte model has inherent limitations in completely representing the intricate structure of human skin. To overcome these limitations, various animal and cell models must be used in different experimental contexts, along with conducting clinical trials for human application.

## 5. Conclusions

In conclusion, our findings demonstrate that PN mitigated AD-like skin inflammation by inducing antioxidant enzyme expression and inhibiting inflammation via modulating the MAPKs–STAT pathway. These results imply that the oral administration of PN holds promise as a prospective anti-inflammatory therapy for AD skin conditions.

## Figures and Tables

**Figure 1 antioxidants-13-00928-f001:**
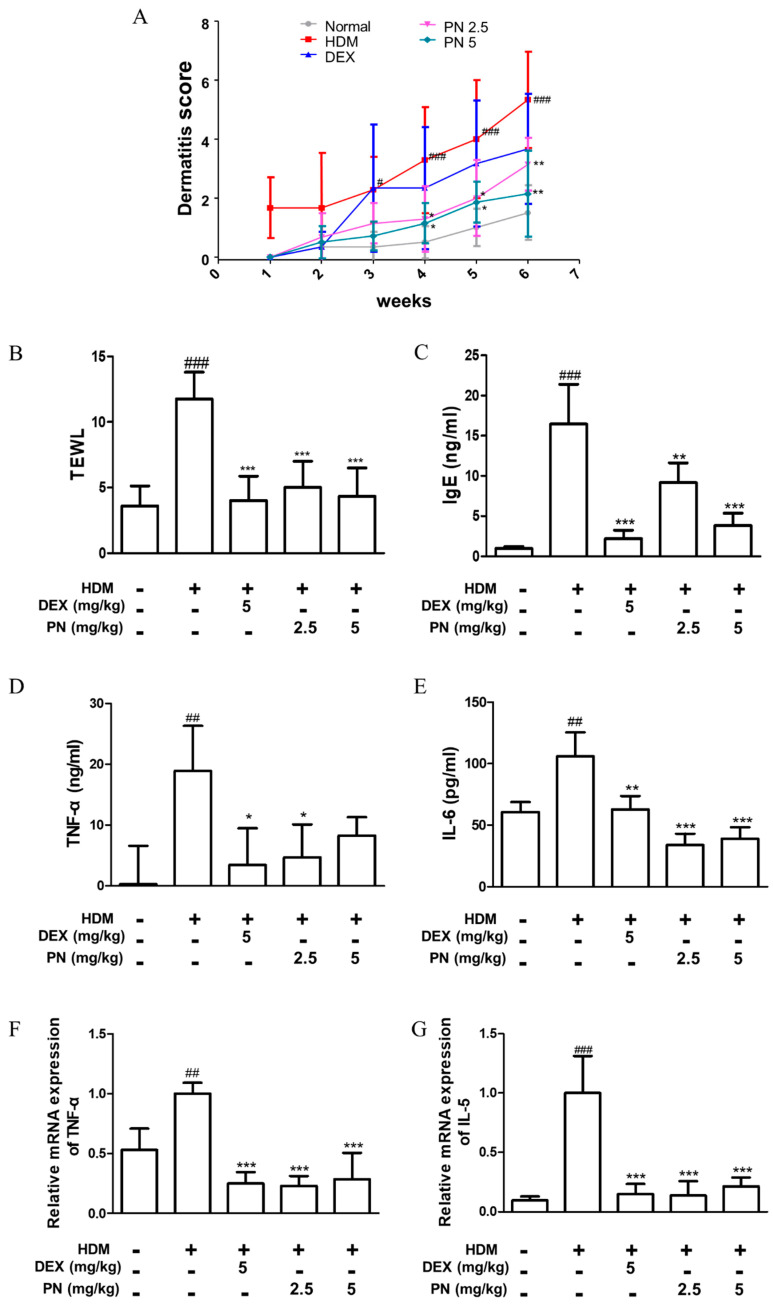
Effects of papain (PN) on the clinical features of the skin of Dfb-induced atopic dermatitis (AD) mice. (**A**) Dermatitis scores were measured once a week for six weeks. The dermatitis score was defined as the sum of scores graded for each symptom. (**B**) Transepidermal water loss (TEWL) was measured by the end of six weeks. (**C**) Serum IgE level was measured using an ELISA kit. (**D**,**E**) Protein levels of TNF-α and IL-6 in dorsal skin tissue were determined using ELISA kits. (**F**,**G**) Total RNA was prepared from the dorsal skin tissue, and mRNA expression levels of TNF-α and IL-6 were determined via RT-qPCR. # *p* < 0.05, ## *p* < 0.01, ### *p* < 0.001 vs. control group; * *p* < 0.05, ** *p* < 0.01, and *** *p* < 0.001 vs. Dfb-induced AD group.

**Figure 2 antioxidants-13-00928-f002:**
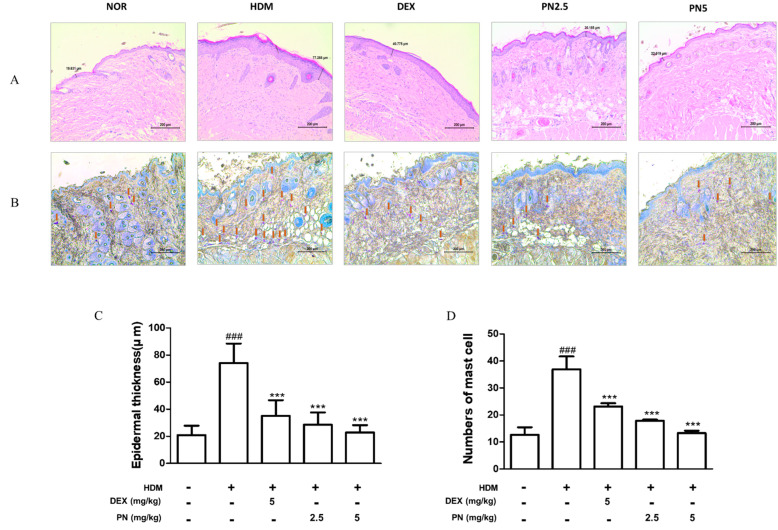
Effects of PN on histological alterations in the skin of Dfb-induced AD mice. (**A**) Hematoxylin and eosin (H&E) staining of skin lesions in AD mice (scale bar = 200 μm). (**B**) Toluidine blue staining of skin lesions in AD mice (scale bar = 200 μm). (**C**) Determination of epidermal thickness. Through the H&E-stained sections, epidermal thickness was measured under a microscope. (**D**) Through the toluidine blue-stained sections, mast cell infiltration was shown as the average count in five fields. This can be confirmed through the brown arrow. Data represent mean ± standard deviation (SD) from three independent experiments. ### *p* < 0.001 vs. control group; *** *p* < 0.001 vs. Dfb-induced AD group.

**Figure 3 antioxidants-13-00928-f003:**
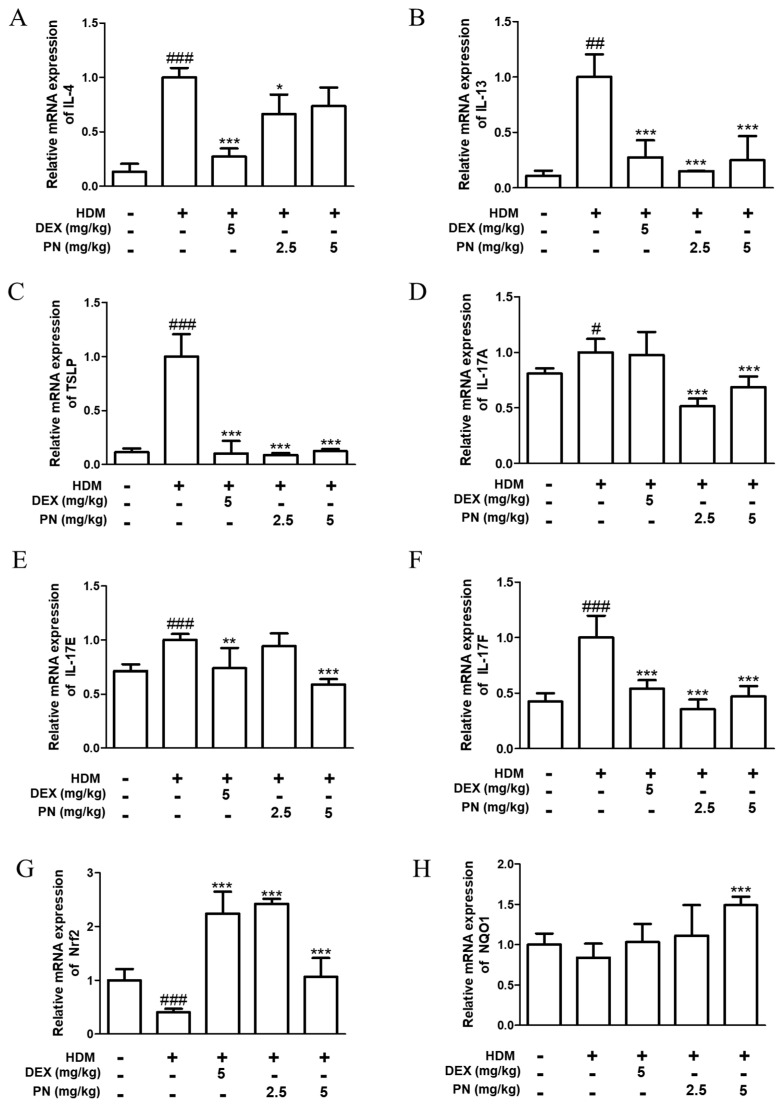
Effects of PN on AD-related cytokines and antioxidant makers in the skin of Dfb-induced AD mice. Total RNA was prepared from the dorsal skin tissue, and the mRNA expression levels of (**A**) IL-4, (**B**) IL-13, (**C**) TSLP, (**D**) IL-17A, (**E**) IL-17E, (**F**) IL-17F, (**G**) NRF-2, and (**H**) NQO1 were determined via RT-qPCR. Data are presented as the mean ± standard deviation from triplicate experiments. # *p* < 0.05, ## *p* < 0.01, ### *p* < 0.001 vs. control group; * *p* < 0.05, ** *p* < 0.01, and *** *p* < 0.001 vs. Dfb-induced AD group.

**Figure 4 antioxidants-13-00928-f004:**
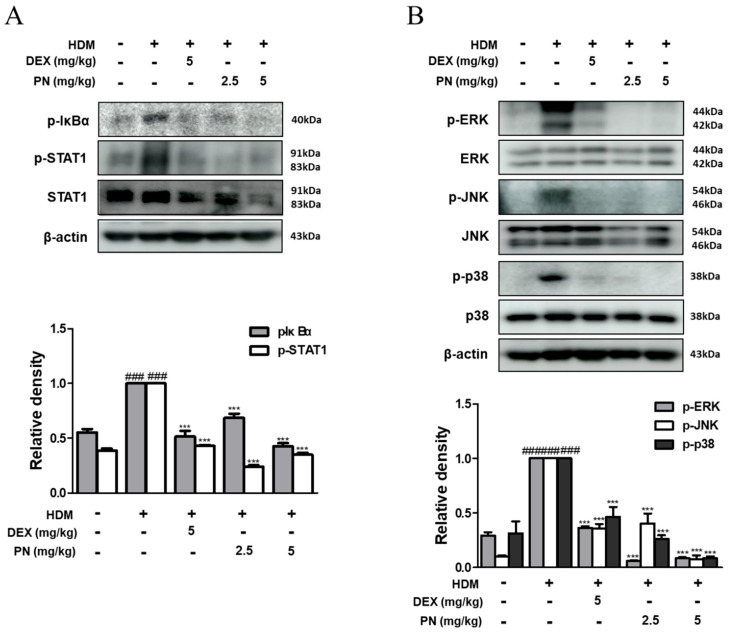
Effects of PN on the activation of MAPKs and STAT1 in the skin of Dfb-induced AD mice. Total proteins were prepared from the dorsal skin, and Western blotting was performed for the determination of (**A**) p-IκBα, p-STAT1, STAT1, (**B**) p-ERK, ERK, p-JNK, JNK, p-p38, and p38 using specific antibodies. β-actin was used as an internal control. Densitometric analysis was determined via Bio-Rad Quantity One^®^ 5.x Software. Data are presented as the mean ± standard deviation from triplicate experiments. ### *p* < 0.001 vs. control group; *** *p* < 0.001 vs. Dfb-induced AD group.

**Figure 5 antioxidants-13-00928-f005:**
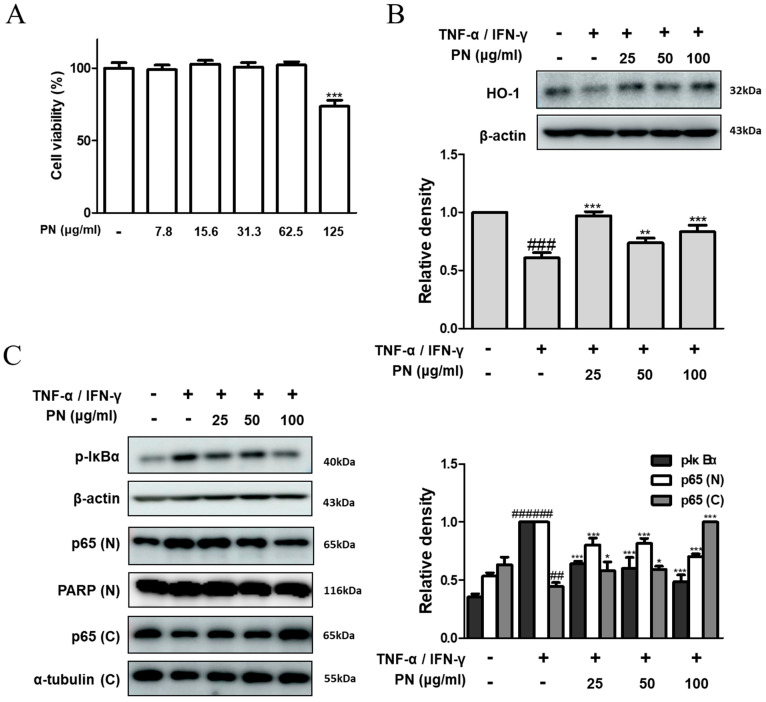
Effects of PN on NF-κB activation in TNF-α/IFN-γ-stimulated HaCaT keratinocytes. (**A**) Viability of HaCaT keratinocytes was measured using MTT assay. *** *p* < 0.001 vs. non-treated group. Total proteins were prepared, and Western blotting was performed for the determination of (**B**) HO-1, (**C**) p-IκBα, and NF-κB p65. β-actin, α-tubulin, and PARP were used as internal control. ## *p* < 0.01, ### *p* < 0.001 vs. control group; * *p* < 0.05, ** *p* < 0.01, and *** *p* < 0.001 vs. TNF-α/IFN-γ-stimulated group.

**Figure 6 antioxidants-13-00928-f006:**
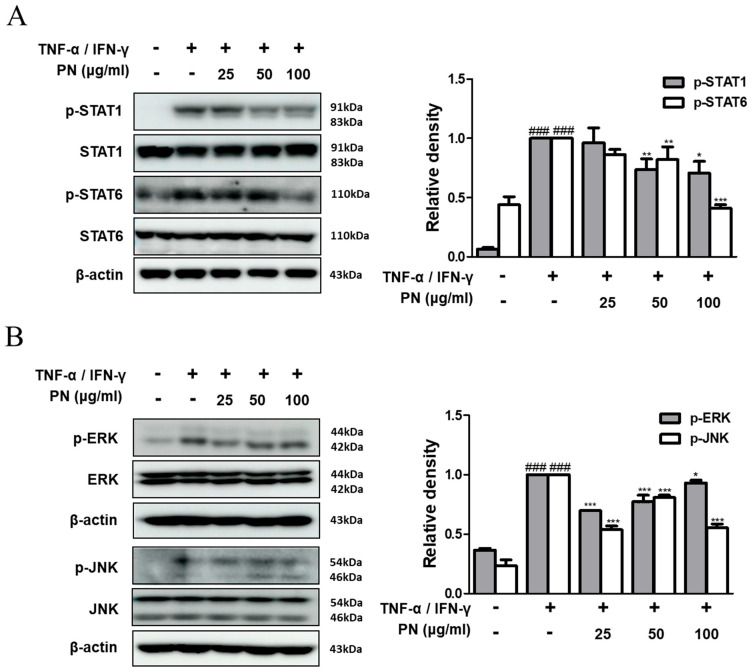
Effects of PN on the activation of MAPKs and STATs in TNF-α/IFN-γ-stimulated HaCaT keratinocytes. Total proteins were prepared, and Western blotting was performed for the determination of (**A**) p-STAT1, STAT1, p-STAT6, STAT6, (**B**) p-ERK, ERK, p-JNK, and JNK. β-actin was used as an internal control. Densitometric analysis was determined via Bio-Rad Quantity One^®^ 5.x Software. Data are presented as the mean ± standard deviation from triplicate experiments. ### *p* < 0.001 vs. control group; * *p* < 0.05, ** *p* < 0.01, and *** *p* < 0.001 vs. TNF-α/IFN-γ-stimulated group.

**Figure 7 antioxidants-13-00928-f007:**
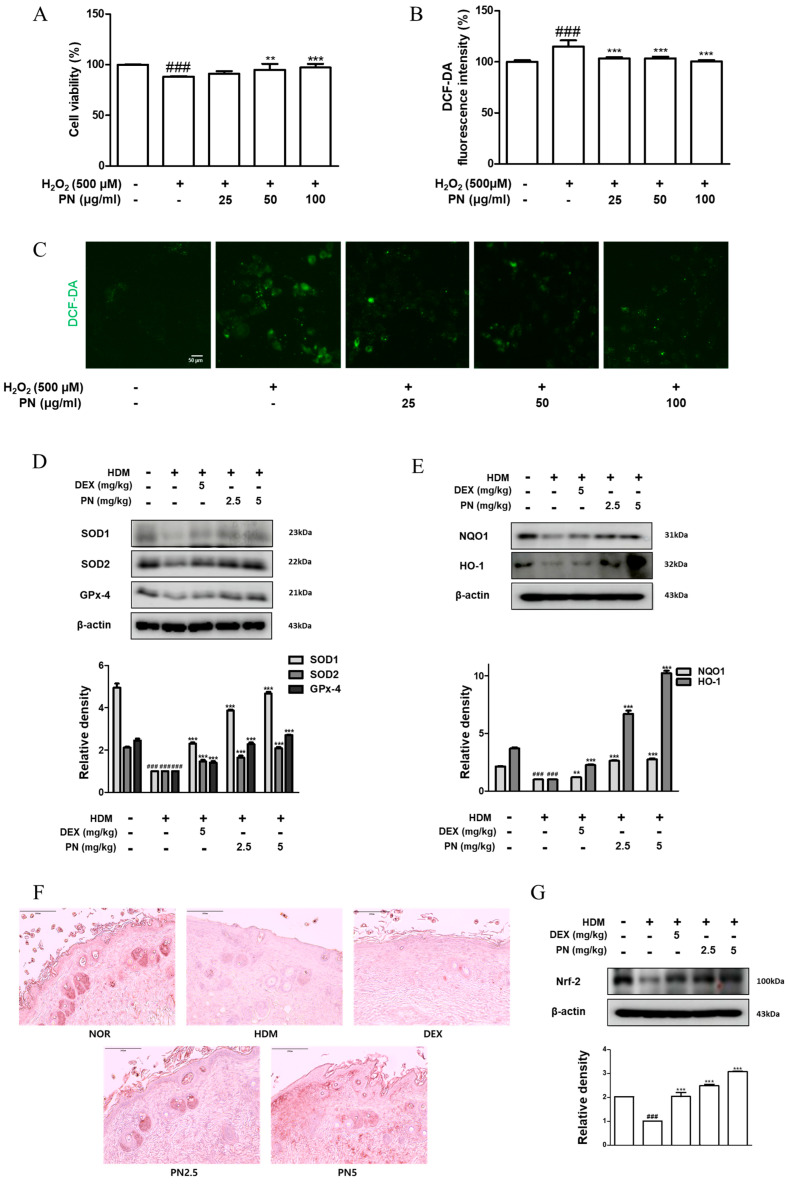
Effects of PN on oxidative damage in H_2_O_2_-induced HaCaT keratinocytes and Dfb-induced AD mice. (**A**) Viability of HaCaT keratinocytes after treatment with PN for 1 h and H_2_O_2_ for 4 h, measured using Cell Counting Kit-8 assay. (**B**) HaCaT cells were labeled with a DCF-DA probe for fluorescent detection, and a representative ROS image was selected (10×). (**C**) Fluorescence microscopic image of ROS induced using H_2_O_2_ after treatment with PN. Total proteins were prepared, and Western blotting was performed for the determination of (**D**) SOD1, SOD2, GPx-4, (**E**) NQO1, HO-1, and (**G**) Nrf2. β-actin was used as internal control. (**F**) Histological sections of dorsal skin tissue were immunohistochemically stained with HO-1 antibody, expressed as brown spots (Scale bar = 200 µm). Data are presented as the mean ± standard deviation from triplicate experiments. ### *p* < 0.001 vs. control group; ** *p* < 0.01, and *** *p* < 0.001 vs. TNF-α/IFN-γ-stimulated group or Dfb-induced AD group.

## Data Availability

Data are contained within the article and Appendix A.

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
