# Peer review of "Papain Suppresses Atopic Skin Inflammation through Anti-Inflammatory Activities Using In Vitro and In Vivo Models"

_antioxidants, 2024, doi:10.3390/antiox13080928_

Round 1

Reviewer 1 Report (Previous Reviewer 1)

The authors have answered correctly to my questions.

The manuscript is suitable for publication

Author Response

Please see the comments in the attachment.

Reviewer 2 Report (Previous Reviewer 2)

Dear Authors

This manuscript is much improved. Here, there are so minor comments. 

1. PN was written in the abstract. Please, write down both full name and abbreviation the first time. 

2. There were several abnormal sentences. Please, moderate English Editing service should be required. 

For example, "The antioxidant efficacy of PN was demonstrated in both in vitro and in vivo models, where PN treatment improved the survival and reduced ROS production in H2O2-damaged HaCaT keratinocytes and enhanced the expression of antioxidant enzymes in Dfb-induced AD mice, primarily through the regulation of nuclear factor erythroid-2-related factor 2 expression."

The abstract should be rephrased for the same reason. 

3. Please, unify the "MAPK" and "MAPKs" 

4. The number of references was too small. Please, add more references. 

5. Please, adjust their contents and Figure 1, 3, and 7

6. Please, enlarge Figure 7C (ROS data), and mark Figure 7F with the arrow. 

7. Please, provide n = 3 in blot original images (not only representative data)

1. PN was written in the abstract. Please, write down both full name and abbreviation the first time. 

2. There were several abnormal sentences. Please, moderate English Editing service should be required. 

For example, "The antioxidant efficacy of PN was demonstrated in both in vitro and in vivo models, where PN treatment improved the survival and reduced ROS production in H2O2-damaged HaCaT keratinocytes and enhanced the expression of antioxidant enzymes in Dfb-induced AD mice, primarily through the regulation of nuclear factor erythroid-2-related factor 2 expression."

The abstract should be rephrased for the same reason. 

3. Please, unify the "MAPK" and "MAPKs" 

4. The number of references was too small. Please, add more references. 

5. Please, adjust their contents and Figure 1, 3, and 7

6. Please, enlarge Figure 7C (ROS data), and mark Figure 7F with the arrow. 

7. Please, provide n = 3 in blot original images (not only representative data)

Author Response

Reviewer 3 Report (Previous Reviewer 3)

Dear Authors,

This is re-submitted version is a good manuscript that dealt with the use of papain to suppress atopic skin inflammation. Such use of PN has merit to be investigated. My concern from the first submission relied on the part of the antioxidant activity that was not an issue, however, I believe this re-submission had improvements and it is in the scope of Antioxidants. Strengths and Limitations section/paragraph was added. It is recommended a minor editing of English. 

The re-submitted version of this manuscript is suitable for acceptance. 

Author Response

This manuscript is a resubmission of an earlier submission. The following is a list of the peer review reports and author responses from that submission.

Round 1

Reviewer 1 Report

This study aimed to investigate the potential effect of papain on skin inflammation in the house dust mite Dermatophagoides 2farinae body (Dfb)-exposed NC/Nga atopic dermatitis (AD) mice and human HaCaT keratinocytes and its underlying mechanisms. The effect of papain on the skin was assessed using histological analysis, transepidermal water loss (TEWL) measurements, enzyme-linked immunosorbent assay western blot analysis, and quantitative reverse transcription-polymerase chain reaction.

The athors  demonstrated that oral administration of papain improved the severity scores of AD-like skin lesions and reduced TEWL, serum immunoglobulin E, and inflammatory cytokine levels in Dfb-in duced AD mice. Papain decreased epidermal thickness and mast cell infiltration through inhibition of AD-related cytokines. Furthermore, papain suppressed the activation of mitogen-activated protein kinase (MAPK) /signal transducer and activator of transcription (STAT) in Dfb-induced AD mice and HaCaT keratinocytes.

The authors conclude that oral administration of papain may act as a potential antioxidant agent by suppressing inflammatory mediators and downregulating the MAPK/STAT pathway, suggesting its potential role in AD pathogenesis.

The manuscript is well written but the figures are not in focus. There is little resolution and it is difficult to know and recognize what the authors want to prove. However, as regards the results, my main concern concerns the histological analyzes carried out and to be carried out. In particular:
a) Have the authors tried to carry out an analysis of the evaluation of the cellular infiltrate in the preparations stained with hematoxylin and eosin?
b) Why did they consider mast cells as cellular markers?
c) Why didn't they also consider other cell types?
d) Why didn't they also consider angiogenesis?
e) Returning to the problem of mast cells, why did the authors limit themselves to considering the results with toluidine blue and did not take into consideration other immunohistochemical or histochemical markers?
f) Are the authors sure that with toluidine blue they are able to mix the responses of the inflammatory state?

Reviewer 2 Report

Dear Authors

This manuscript is relatively well-organized.  

Please, add a representative cartoon as a Figure. 7 regarding the effects of papain on AD mice including molecular mechanism. 

1. Please, write down MAPKs, not MAPK. 

2. The number of references was too small. Please, add more references above at least 40. 

3. In the part of M&M, please, write down the gene name of primers and cytokine assays in detail. 

4. Please, do not include Supplementary Figure 1 in this manuscript, and prepare the supplementary article.

5. Please, omit the data of beta-actin in Figure 4A.

Reviewer 3 Report

Dear Authors,

This is a good manuscript dealing with the use of papain to suppress atopic skin inflammation. Such use of PN has merit to be investigated. My concern relies on the part of the antioxidant activity that was not an issue in this manuscript, thus, this content may not be in the scope of Antioxidants. Another point, please add a Strengths and Limitations section/paragraph involving the use of animal model and cell culture. Perspectives would be welcome. 

My concern relies on the part of the antioxidant activity that was not an issue in this manuscript, thus, this content may not be in the scope of Antioxidants. Another point, please add a Strengths and Limitations section/paragraph involving the use of animal model and cell culture. Perspectives would be welcome.